# Transferable Adversarial Robustness for Categorical Data via Universal Robust Embeddings

**Klim Kireev, Maksym Andriushchenko, Carmela Troncoso, Nicolas Flammarion**
EPFL

## Abstract

Research on adversarial robustness is primarily focused on image and text data. Yet, many scenarios in which lack of robustness can result in serious risks, such as fraud detection, medical diagnosis, or recommender systems often do not rely on images or text but instead on *tabular data*. Adversarial robustness in tabular data poses two serious challenges. First, tabular datasets often contain *categorical features*, and therefore cannot be tackled directly with existing optimization procedures. Second, in the tabular domain, algorithms that are not based on deep networks are widely used and offer great performance, but algorithms to enhance robustness are tailored to neural networks (e.g. adversarial training). The code for our method is publicly available at https://github.com/spring-epfl/Transferable-Cat-Robustness.

In this paper, we tackle both challenges. We present a method that allows us to train adversarially robust deep networks for tabular data and to transfer this robustness to other classifiers via *universal robust embeddings* tailored to categorical data. These embeddings, created using a bilevel alternating minimization framework, can be transferred to boosted trees or random forests making them robust *without the need for adversarial training* while preserving their high accuracy on tabular data. We show that our methods outperform existing techniques within a practical threat model suitable for tabular data. The code for our method is publicly available [1].

## 1 Introduction

Works on adversarial machine learning primarily focus on deep networks and are mostly evaluated on image data. Apruzzese et al. (2022) estimate that approximately 70%-80% of works in the literature fall in this category. Yet, a large number of high-stake tasks across fields like medical diagnosis (Shehab et al., 2022), fraud detection (Altman, 2021), click-through rate prediction (Yang & Zhai, 2022), or credit scoring (Shi et al., 2022) neither only rely on deep networks nor operate on images (Grinsztajn et al., 2022). These tasks often involve many discrete categorical features (e.g., country, email, day of the week), and the predominant models used are discrete tree-based (e.g., boosted tree ensembles, random forests). These two characteristics raise a number of challenges when trying to achieve robustness using previous works which focus on continuous features and continuous models (Goodfellow et al., 2014; Madry et al., 2018).

**Accurate modelling of adversarial capability.** The de-facto standard in adversarial robustness evaluation (Croce et al., 2020) is to model robustness to perturbations bounded in some $\ell_p$ ball, mostly in the image domain. This approach, however, was shown to not accurately represent the capabilities of a real adversary in other domains (Kireev et al., 2022; Apruzzese et al., 2022). Instead, a realistic threat model would constrain the adversary with respect to their *financial* capabilities. This can be achieved by associating a financial cost with every input feature transformation, limiting the adversary to perform transformations within their total financial budget. Such a constraint is common

---

[1] https://github.com/spring-epfl/Transferable-Cat-Robustness

for computer security problems as it ties security, in this case, robustness, to real-world limitations. The inaccuracy of threat models in the literature translates on a lack of well-motivated benchmarks for robustness research on tabular data, unlike for image-oriented vision tasks (Croce et al., 2020; Koh et al., 2020).

**Accounting for discrete categorical features.** Tabular data is usually heterogeneous and often includes *categorical* features which can be manipulated by an adversary in a non-uniform way, depending on the characteristics of the real-world concept they represent. For example, buying an email address is not the same as buying a credit card or moving to a different city. Moreover, for a given feature, not all transformations have equal cost or are even possible: for example, one can easily change their email address to `*@gmail.com`, while changing the domain name to `*@rr.com` can be impossible, since this domain name is unavailable for new users. Hence, the definitions of perturbation sets describing the capabilities of potential adversaries should support complex and heterogeneous constraints.

**Robustness for models other than neural networks.** Gradient-based attacks based on projected gradient descent provide a simple and efficient method for crafting adversarial examples. They are effectively used for *adversarial training*, which became a de-facto standard defence against adversarial perturbations (Madry et al., 2018). Albeit defences and attacks are proposed for decision tree models, they often employ combinatorial methods and can be very inefficient time-wise (Kantchelian et al., 2016; Calzavara et al., 2020; Kireev et al., 2022). However, in tasks involving tabular data, these models must be prioritized as they are widely used because they can provide superior performance on some tabular datasets than neural networks (Grinsztajn et al., 2022).

**Contributions.** Our contributions address the challenges outlined above as follows:

- We propose a practical adversarial training algorithm supporting complex and heterogeneous constraints for categorical data that can accurately reflect financial costs for the adversary. Our training algorithm is based on the continuous relaxation of a discrete optimization problem and employs approaches from projections onto an intersection of convex sets.

- We propose a method to generate *universal robust embeddings*, which can be used for transferring robustness from neural networks to other types of machine learning models such as decision trees or random forests.

- We use existing datasets to build the first benchmark that allows us to evaluate robustness for tabular tasks in which the adversary is constrained by financial capabilities.

- Using the proposed benchmark, we empirically show that our proposed methods provide significantly better robustness than previous works.

## 2 Related works

Here we discuss the most relevant references related to the topics outlined in the introduction.

**Gradient-based adversarial training.** Adversarial training is the key algorithm for making neural networks robust to standard $\ell_p$-bounded adversarial examples. Szegedy et al. (2013) demonstrate that modern deep networks are susceptible to adversarial examples that can be easily found via gradient descent. Madry et al. (2018) perform successful adversarial training for deep networks where each iteration of training uses projected gradient descent to find approximately optimal adversarial examples. Related to our method of universal first-layer embeddings, Bakiskan et al. (2022) perform *partial* adversarial training (i.e., training the whole network adversarially/normally and then reinitializing some layer and training them in the opposite way) which is somewhat, showing that earlier layers are more important for robustness. On a related note, Zhang et al. (2019) do adversarial training on the whole network but do multiple forward-backward passes on the first layer with the goal of speeding up adversarial training. Yang et al. (2022) use metric learning with adversarial training to produce word embeddings robust to word-level adversarial attacks which can be used for downstream tasks. Dong et al. (2021) also produce robust word embeddings via a smart relaxation of the underlying discrete optimization problem. We take inspiration from this line of work when we adapt adversarial training for neural networks to categorical features and transfer the first-layer embeddings to tree-based models.

**Robustness of tree-based models.** Tree-based models such as XGBoost (Chen & Guestrin, 2016) are widely used in practice but not amenable to gradient-based adversarial training. Most of the works on adversarial robustness for trees focus on $\ell_\infty$ adversarial examples since they are easier to work with due to the coordinate-aligned structure of decision trees. Kantchelian et al. (2016) is the first algorithm for training robust tree ensembles which are trained on a pool of adversarial examples updated on every iteration of boosting. This approach is refined by approximately solving the associated min-max problem for $\ell_\infty$ robustness in Chen et al. (2019) and by minimizing an upper bound on the robust loss in Andriushchenko & Hein (2019) on each split of every decision tree in the ensemble. Chen et al. (2021) extend the $\ell_\infty$ approach of Chen et al. (2019) to arbitrary *box* constraints and apply it to a cost-aware threat model on continuous features. Only few works tackle non-$\ell_\infty$ robust training of trees since their coordinate-aligned structure is not conducive to other norms. Most related work to ours is Wang et al. (2020) extend the upper bounding approach of Andriushchenko & Hein (2019) for arbitrary $\ell_p$-norms. However, they report that $\ell_\infty$ robust training in most cases works similarly to $\ell_1$ robust training, though in a few cases $\ell_1$ adversarial training yields better performance. Moreover, they do not use categorical features which are typically present in tabular data which is the focus of our work.

**Threat modeling for tabular data.** Many prior works do not consider a realistic threat model and adversarial capabilities. First, popular benchmarks like Madry et al. (2018); Croce et al. (2020) assume that the adversary has an *equal* budget for perturbing each feature which is clearly not realistic for tabular data. To fix this, Chen et al. (2021) propose to consider different perturbation costs for different features. Kireev et al. (2022) argue that, as imperceptibility and semantic similarity are not necessarily meaningful considerations for tabular datasets, these costs must be based on financial constraints – i.e., how much money it costs for the adversary to execute an attack for a *particular* example. The importance of monetary costs was corroborated by studies involving practitioners dealing with adversarial machine learning in real applications (Apruzzese et al., 2022; Grosse et al., 2023). Second, popular benchmarks treat equally changes from the correct to any other class, but in practice the adversary is typically interested only in a *specific* class change, e.g., modifying a fraudulent transaction to be classified as non-fraudulent. To address this shortcoming, past works (Zhang & Evans, 2019; Shen et al., 2022) propose class-sensitive robustness formulations where an unequal cost can be assigned to every pair of classes that can be changed by the adversary. As a result, this type of adversarial training can achieve a better robustness-accuracy tradeoff against a cost-sensitive adversary. We take into account all these works and consider a cost-based threat model where we realistically model the costs and classes which are allowed to be changed for the particular datasets we use.

## 3 Definitions

**Input domain and Model.** The input domain's *feature space* $\mathbb{X}$ is composed of $m$ features: $\mathbb{X} \subseteq \mathbb{X}_1 \times \mathbb{X}_2 \times \cdots \times \mathbb{X}_m$. We denote as $x_i$ the value of the $i$-th feature of $x \in \mathbb{X}$. Features $x_i$ can be categorical, ordinal, or numeric, and we define tabular data as data consisting of these three types of features in any proportion. Categorical features are features $x_i$ such that $\mathbb{X}_i$ is a finite set of size $|\mathbb{X}_i| = t_i$, i.e., $t_i$ is the number of possible values that can take $x_i$. We denote by $t = \sum_{i=1}^m t_i$. We also denote as $x_i^j$, the $j$-th value of the feature $x_i$. We further assume that each example $x \in \mathbb{X}$ is associated with a binary label $y \in \{0, 1\}$. Finally, we consider a general learning model with parameters $\theta$ and output $\eta(\theta, x)$. The model we consider in our framework can take on two forms: differentiable, such as a neural network, or non-differentiable, like a tree-based model.

**Perturbation set.** Following the principles described by Apruzzese et al. (2022); Kireev et al. (2022), we make the assumption that a realistic adversary is constrained by financial limitations. Specifically, we denote the cost of modifying the feature $x_i$ to $x_i'$ as $c_i(x_i, x_i')$. For categorical features with a finite set $\mathbb{X}_i$, the cost function $c_i(x_i, x_i')$ can be conveniently represented using a *cost matrix* $C_i \in \mathbb{R}_{\geq 0}^{t_i \times t_i}$. The cost of changing the feature $x_i$ from the value $j$ to the value $k$ is given by

$$c_i(x_i^j, x_i^k) = C_i^{jk}.$$

If such transformation is impossible, $C_i^{jk}$ is set to $\infty$. In a realistic threat model, any cost matrix $C_i$ is possible, and costs need not be symmetric. The only reasonable assumption is that $C_i^{jj} = 0$, indicating no cost for staying at the same value. See realistic cost matrix examples in the Appendix A.

We assume that the cost of modifying features is additive: the total cost of crafting an adversarial example $x'$ from an example $x$ can be expressed as:

$$c(x, x') = \sum_{i=1}^{m} c_i(x_i, x_i').\qquad(1)$$

**Encoding and Embeddings.** Categorical features are typically preprocessed by encoding each value $x_i$ using a one-hot encoding vector $\overline{x}_i$. For instance, if a categorical feature $x_i$ can take four possible values $\{1, 2, 3, 4\}$ and $x_i = 2$, it is represented as $\overline{x}_i = (0, 1, 0, 0)^\top$. We can then represent the feature-wise cost function as

$$c_i(x_i, x_i') = \|w_i \odot (\overline{x}_i - \overline{x}_i')\|_1 = l_{1,w_i}(\overline{x}_i, \overline{x}_i'),\qquad(2)$$

where $l_{1,w}$ denotes the weighted $l_1$ norm and $w_i = C_i\overline{x}_i$ is the costs of transforming $x_i$ to any other possible value in $\mathbb{X}_i$. Then according to Equation (1), the per sample cost function is:

$$c(x, x') = \|w_i \odot (\overline{x} - \overline{x}')\|_1 = l_{1,w}(\overline{x}, \overline{x}'),\qquad(3)$$

where the vectors $w, \overline{x}, \overline{x}'$ are the contenation of the vectors $w_i, \overline{x}_i, \overline{x}_i'$ respectively for $\forall i : 0 \leq i \leq m$.

The one-hot-encoded representation may not be optimal for subsequent processing due to high dimensionality or data sparsity. We instead use *embeddings* and replace the categorical features with their corresponding embedding vector $\phi(x) \in R^n$. The model can then be represented by a function $f$ of the features as

$$\eta(\theta, x) = f(\theta, \phi(x)).$$

The embedding function $\phi$ is composed of embedding functions $\phi_i$ for the subvector $x_i$ as $\phi(x) = (\phi_i(x_i))_{i=1}^m$. The embedding function $\phi_i$ of the $i$-th feature can always be represented by a matrix $Q_i$ since $x_i$ takes discrete values. The columns of this matrix are *embedding vectors* which satisfy

$$\phi_i(x_i) = Q_i\overline{x}_i.$$

Therefore, we parametrize the embedding function $\phi(x, Q)$ with a family of $m$ matrices $Q = (Q_i)_{i=1}^m$.

**Numerical features.** For numerical features, we use a binning procedure which is a common practice in tabular machine learning. This technique facilitates the use of decision-tree-based classifiers as they naturally handle binned data (Chen & Guestrin, 2016; Ke et al., 2017b). Binning transforms numerical features into categorical ones, allowing a unique treatment of all the features. It is worth noting that for differentiable classifiers, the approach by Kireev et al. (2022) can enable us to directly use numerical features without binning.

## 4  Adversarial training

In this section, we outline our adversarial training procedure. When adapted to our specific setup, the adversarial training problem (Madry et al., 2018) can be formulated as:

$$\min_{\theta, Q} \mathbb{E}_{x,y \in D} \max_{c(x,x') \leq \varepsilon} \ell((f(\phi(x', Q)), \theta), y).\qquad(4)$$

Such model would be robust to adversaries that can modify the value of any feature $x_i$, as long as the total cost of the modification is less than $\varepsilon$. While this formulation perfectly formalizes our final goal, direct optimization of this objective is infeasible in practice:

1. Optimizing directly with categorical features can be computationally demanding due to the need for discrete optimization algorithms. In our evaluation, we employ a graph-search-based procedure developed by Kireev et al. (2022). However, finding a single example using this algorithm can take between 1 to 10 seconds, making it impractical for multi-epoch training even on medium-sized datasets (approximately 100K samples).

2. There is currently no existing algorithm in the literature that enables us to operate within this cost-based objective for decision-tree based classifiers.

**Algorithm 1** Cat-PGD. Relaxed projected gradient descent for categorical data.

---

**Input:** Data point $\tilde{x}, y$, Attack rate $\alpha$, Cost bound $\varepsilon$, Cost matrices $C$, $PGD\_steps$, $D\_steps$
**Output:** Adversarial sample $\tilde{x}'$.
$\delta := 0$
**for** $i = 1$ **to** $PGD\_steps$ **do**
   $\nabla := \nabla_\delta \ell(f(Q(\tilde{x} + \delta)), y))$
   $\delta := \delta + \alpha \nabla$
   $p := 0$
   $q := 0$
   **for** $i = 1$ **to** $D\_steps$ **do**
      $z := \Pi_{simplices}(\tilde{x}, \delta + p)$
      $p := \delta + p - z$
      $\delta := \Pi_{cost}(z + q, C)$
      $q := z + q - \delta$
   **end for**
**end for**
$\tilde{x}' := \tilde{x} + \delta$

---

### 4.1 Adversarial training for differentiable models

We begin by examining the scenario where our model $f$ is a deep neural network. In this case, we are constrained solely by the first restriction. To address this constraint, we employ a *relaxation technique*: instead of working with the discrete set of feature vectors, we consider the convex hull of their one-hot encoding vectors. For each feature, this relaxation allows us to replace the optimization over $x'_i \in \mathbb{X}_i$ with an optimization over $\tilde{x}'_i \in \mathbb{R}^{t_i}_{\geq 0}$. More precisely, we first replace the feature space $\mathbb{X}_i$ by the set $\{\overline{x}'_i \in \{0,1\}^{t_i}, \sum_j \overline{x}'^j_i = 1\}$ using the one-hot encoding vectors. We then relax this discrete set into $\{\tilde{x}'_i \in \mathbb{R}^{t_i}_{\geq 0}, \sum_j \tilde{x}'^j_i = 1\}$. The original constraint $\{x' \in \mathbb{X}, c(x, x') \leq \varepsilon\}$ is therefore relaxed as

$$\{\tilde{x}'_i \in \mathbb{R}^{t_i}_{\geq 0}, \sum_j \tilde{x}'^j_i = 1, l_{1,w}(\overline{x}, \tilde{x}') \leq \varepsilon \text{ and } \tilde{x}' = (\tilde{x}_i)^m_{i=1}\}.$$

The set obtained corresponds to the convex hull of the one-hot encoded vectors, providing the tightest convex relaxation of the initial discrete set of vectors $\overline{x}$. In contrast, the relaxation proposed by Kireev et al. (2022) relaxes $\mathbb{X}_i$ to $\mathbb{R}^{t_i}_{\geq 0}$, leading to significantly worse performance as shown in Section 5.

The adversarial training objective for the neural networks then becomes:

$$\min_{\theta, Q} \mathbb{E}_{x, y \in D} \max_{\substack{l_{1,w}(\overline{x}, \tilde{x}') \leq \varepsilon \\ \|\tilde{x}'_i\|_1 = 1, \tilde{x}'_i \in \mathbb{R}^{t_i}_{\geq 0} \text{ for } 1 \leq i \leq m}} \ell(f(Q\tilde{x}', \theta), y). \tag{5}$$

In order to perform the inner maximization, we generate adversarial examples using projected gradient descent. The projection is computed using the Dykstra projection algorithm (Boyle & Dykstra, 1986) which enables us to project onto the intersection of multiple constraints. In each iteration of the projection algorithm, we first project onto the convex hull of each categorical feature ($\Pi_{simplices}$), followed by a projection onto a weighted $l_1$ ball ($\Pi_{cost}$). This dual projection approach allows us to generate perturbations that simultaneously respect the cost constraints of the adversary and stay inside the convex hull of original categorical features. We refer to this method as Cat-PGD, and its formal description can be found in Algorithm 1.

### 4.2 Bilevel alternating minimization for universal robust embeddings

While the technique described above is not directly applicable to non-differentiable models like decision trees, we can still leverage the strengths of both neural networks and tree-based models. By transferring the learnt robust embeddings from the *first* layer of the neural network to the decision tree classifier, we can potentially combine the robustness of the neural network with the accuracy of boosted decision trees.

---

**Algorithm 2** Bilevel alternating minimization for universal robust embeddings

---
**Input:** Set of training samples $X, y$, cost bound $\varepsilon$, $\theta\_steps$, $Q\_steps$, $PGD\_steps$, $N\_iters$.
**Output:** Embeddings $Q_{N\_iters}$.
Randomly initialize $Q, \theta$.
**for** $i = 1$ **to** $N\_iters$ **do**
   $\theta_i^1 := \theta_i, Q_i^1 := Q_i$
   **for** $j = 1$ **to** $\theta\_steps$ **do**
      $\theta_i^{j+1} := \theta_j - \alpha \nabla_\theta \ell(f(Q_i \overline{x}, \theta_i^j), y)$
   **end for**
   $\theta_i := \theta_i^{\theta\_steps}$
   **for** $j = 1$ **to** $Q\_steps$ **do**
      $\tilde{x}' := \mathbf{Attack}(\overline{x}, y, PGD\_steps, Q_i^j, \theta_i)$
      $Q_i^{j+1} := Q_i^j - \beta \nabla_Q \ell(f(Q_i^j \tilde{x}', \theta_i), y)$
   **end for**
   $\theta_{i+1} := \theta_i^{\theta\_steps}, Q_{i+1} := Q_i^{Q\_steps}$
**end for**

---

**Difference between input and output embeddings.** Using embeddings obtained from the last layer of a neural network is a common approach in machine learning (Zhuang et al., 2019). However, in our study, we prefer to train and utilize embeddings from the *first layer* where the full information about the original features is still preserved. This choice is motivated by the superior performance of decision-tree-based classifiers over neural networks on tabular data in certain tasks. By using first layer embeddings, we avoid a potential loss of information (unless the embedding matrix $Q$ has exactly identical rows which are unlikely in practice) that would occur if we were to use embeddings from the final layer.

To show this effect we ran a simple experiment where we compare first and last layer embeddings as an input for a random forest classifier. We report the results in Appendix C.1.

**Step 1: bilevel alternating minimization framework.** A natural approach is to use standard adversarial training to produce robust first layer embeddings. However, we noticed that this method is not effective in producing optimal first layer embeddings. Instead, we specifically produce robust first layer embeddings and ignore the robustness property for the rest of the layers. This idea leads to a new objective that we propose:

**Bilevel minimization**: $$\min_Q \; \mathbb{E}_{x,y \in D} \; \max_{\substack{l_{1,w}(\overline{x}, \tilde{x}') \leq \varepsilon \\ \|\tilde{x}_i'\|_1 = 1, \tilde{x}_i' \in \mathbb{R}_{\geq 0}^{t_i} \text{ for } 1 \leq i \leq m}} \ell\Big( f\Big(Q\tilde{x}, \arg\min_\theta \; \mathbb{E}_{x,y \in D} \ell\big(f(Q\tilde{x}', \theta), y\big)\Big), y\Big).$$

(6)

This optimization problem can be seen as the relaxation of the original adversarial training objective. This relaxation upper bounds the original objective because we robustly optimize only over $Q$ and not jointly over $Q$ and $\theta$. If we disregard the additional inner maximization, it is a classical *bilevel optimization problem*. Such problem can be solved using alternating gradient descent (Ghadimi & Wang, 2018). To optimize this objective for neural networks on large tabular datasets, we propose to use *stochastic* gradient descent. We alternate between $Q\_steps$ for the inner minimum and $\theta\_steps$ for the outer minimum. At each step, we run projected gradient descent for $PGD\_steps$ using the relaxation described in Eq. 5. We detail this approach in Algorithm 2.

**Step 2: embedding merging algorithm.** We observed that directly transferring the embeddings $Q$ has little effect in terms of improving the robustness of decision trees (Appendix C.1). This is expected since even if two embeddings are very close to each other, a decision tree can still generate a split between them. To address this issue and provide decision trees with information about the relationships between embeddings, we propose a merging algorithm outlined in Algorithm 3. The main idea of the algorithm is to merge embeddings in $Q$ that are very close to each other and therefore pass the information about distances between them to the decision tree.

**Step 3: standard training of trees using universal robust embeddings.** As the final step, we use standard tree training techniques with the *merged embeddings* $Q'$. This approach allows us to avoid the need for custom algorithms to solve the relaxed problem described in Eq. 5. Instead, we can

---
**Algorithm 3** Embeddings merging algorithm
---
**Input:** Embeddings $Q$, percentile $p$
**Output:** Embeddings $Q'$.
1) $D := []$
2) For each $q_j, q_k \in Q_i$ compute $d_{jk} = ||q_j - q_k||_2$ and put it to $D$.
3) Sort $D$
4) For a given percentile, compute threshold $t$, s.t. all $d > t$ belong to the percentile
5) Cluster $Q$ using $t$ as a maximum distance between two points in one cluster
6) In each cluster compute average embedding vector and put it into $Q'$
---

leverage highly optimized libraries such as XGBoost or LightGBM (Chen & Guestrin, 2016; Ke et al., 2017b). The motivation behind this multi-step approach is to combine the benefits of gradient-based optimization for generating universal robust embeddings with the accuracy of tree-based models specifically designed for tabular data with categorical features.

# 5    Evaluation

In this section we evaluate the performance of our methods.

**Models.** We use three 'classic' classifiers widely used for tabular data tasks: RandomForest (Liaw & Wiener, 2002), LGBM (Ke et al., 2017a), and Gradient Boosted Stumps. Both for adversarial training and for robust embeddings used in this section, we use TabNetArik & Pfister (2019) which is a popular choice for tabular data. Additionally, we also show the same trend with FT-Transformer in Appendix C. The model hyperparameters are listed in Appendix B.

**Attack.** For evaluation, we use the graph-search based attack described by Kireev et al. (2022). This attack is model-agnostic and can generate adversarial examples for both discrete and continuous data, and is thus ideal for our tabular data task.

**Comparison to previous work.** In our evaluation, we compare our method with two previous proposals: the method of Wang et al. (2020), where the authors propose a verified decision trees robustness algorithm for $l_1$ bounded perturbations; and the method of Kireev et al. (2022), which considers financial costs of the adversary, but using a different cost model than us and weaker relaxation.

## 5.1    Tabular data robustness evaluation benchmark

There is no consistent way to evaluate adversarial robustness for tabular data in the literature. Different works use different datasets, usually without providing a justification neither for the incentive of the adversary nor for the perturbation set. For example, Wang et al. (2020) and Chen et al. (2021) use the breast cancer dataset, where participants would not have incentives to game the classifier, as it would reflect poorly on their health. Even if they had an incentive, it is hard to find plausible transformation methods as all features relate to breast images taken by doctors. To solve this issue, we build our own benchmark. We select datasets according to the following criteria:

- Data include both numerical and categorical features, with a financial interpretation so that it is possible to model the adversarial manipulations and their cost.

- Tasks on the datasets have a financial meaning, and thus evaluating robustness requires a cost-aware adversary.

- Size is large enough to accurately represent the underlying distribution of the data and avoid overfitting. Besides that, it should enable the training of complex models such as neural networks.

It is worth mentioning that the financial requirement and the degree of sensitivity of data are highly correlated. This kind of data is usually not public and even hardly accessible due to privacy reasons. Moreover, the datasets are often imbalanced. Some of these issues can be solved during preprocessing stage (e.g., by balancing the dataset), but privacy requirements imply that benchmarks must mostly rely on anonymized or synthetic versions of a dataset.

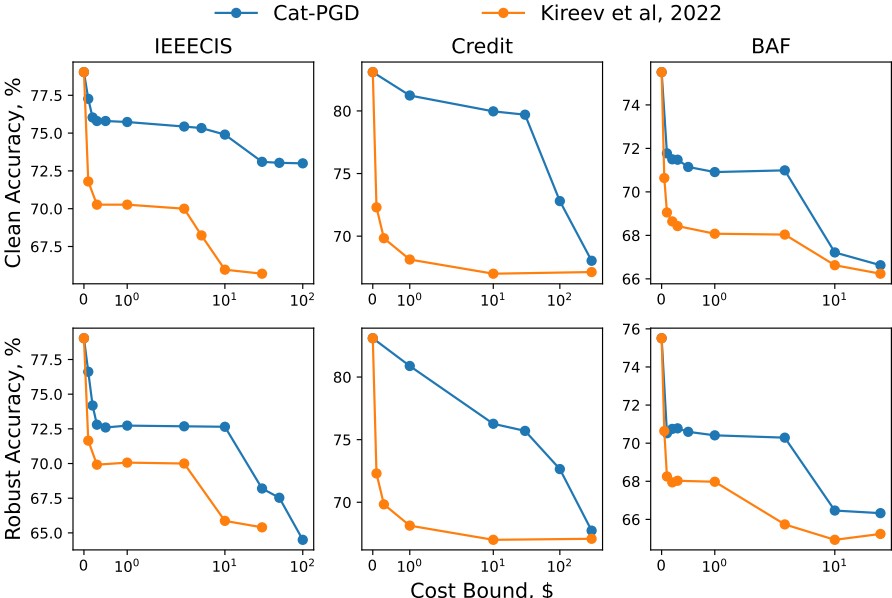

Figure 1: **Clean and robust model accuracy for cost-aware adversarial training.** Each point represents a TabNet model trained with an assumption of an adversary's budget of $\varepsilon$ \$ attacked by the adversary with that budget. Our method outperforms Kireev et al. (2022) in all setups.

**Datasets.** We select three publicly-available datasets that fit the criteria above. All three datasets are related to real-world financial problems where robustness can be crucially important. For each dataset, we select adversarial capabilities for which we can outline a plausible modification methodology, and we can assign a plausible cost for this transformation.

- `IEEECIS`. The `IEEECIS` fraud detection dataset (Kaggle, 2019) contains information about around 600K financial transactions. The task is to predict whether a transaction is fraudulent or benign.
- `BAF`. The Bank Account Fraud dataset was proposed in NeurIPS 2022 by Jesus et al. (2022) to evaluate different properties of ML algorithms for tabular data. The task is to predict if a given credit application is a fraud. It contains 1M entries for credit approval applications with different categorical and numerical features related to fraud detection.
- `Credit`. The credit card transaction dataset (Altman, 2021) contains around 20M simulated card transactions, mainly describing purchases of goods. The authors simulate a "virtual world" with a "virtual population" and claim that the resulting distribution is close to a real banking private dataset, which they cannot distribute.

All datasets were balanced for our experiments. The features for the adversarial perturbations along with the corresponding cost model are described in Appendix A.

### 5.2 Results

**Cost-aware Adversarial Training.** We follow a standard evaluation procedure: we first perform adversarial training of the TabNet models using different cost bounds $\varepsilon$, representing different financial constraints for the adversary. Then, we evaluate these models' robustness using the attack by Kireev et al. (2022) configured for the same cost adversarial constraint as used during the training. The details about our training hyperparameters are listed in the Appendix B.

We show the resulting clean and robust performance in Figure 1. Our method outperforms the baseline in both metrics. The better performance can be attributed to a better representation of the threat model. For example, let us consider the email address feature in `IEEECIS`. The cost to change this feature varies from 0.12\$ (e.g., hotmail.com) to $\infty$ (for emails that are no longer available). It

Table 1: **Universal robust embedding evaluation.** We report clean and robust accuracy (in percentage) for Light Gradient Boosting (LGBM), Random Forest (RF), Gradient Boosted Stumps (GBS), and CatBoost (CB). We indicate the robustness technique applied as a suffix: -R for robust embeddings, -W for the training proposed by Wang et al. (2020), and -C for method in Chen et al. (2019). Models fed with robust embeddings have higher robustness and outperform both clean and $l_1$-trained models (Wang et al., 2020).

| Dataset | RF | RF-R | RF-C | LGBM | LGBM-R | GBS | GBS-R | GBS-W | CB | CB-R |
|---------|------|------|------|------|--------|------|-------|-------|------|------|
| IEEECIS |
| Clean | 83.6 | 81.0 | 69.0 | 81.2 | 79.3 | 66.9 | 66.3 | 52.4 | **76.5** | 76.1 |
| Robust | 48.0 | **81.0** | 69.0 | 53.9 | 78.5 | 44.7 | **66.3** | 11.1 | 51.1 | **72.0** |
| BAF |
| Clean | 72.3 | 65.8 | 61.3 | 74.1 | 68.1 | 74.1 | 68.1 | 64.8 | 74.4 | 67.2 |
| Robust | 42.8 | **65.8** | 61.3 | 49.2 | **67.5** | 46.8 | **67.7** | 33.5 | 48.2 | **67.2** |
| Credit |
| Clean | 78.0 | 73.4 | - | 83.1 | 79.6 | 82.1 | 80.6 | 61.7 | 83.3 | 79.7 |
| Robust | 55.2 | **66.7** | - | 69.9 | **72.5** | 70.9 | **71.4** | 61.3 | 71.7 | 71.9 |

is unlikely that an adversary uses an expensive email (sometimes even impossible as in the case of unavailable domains), and therefore such domains are useful for classification as they indicate non-fraudulent transactions. On the other hand, an adversary can easily buy a *gmail.com* address even when their budget $\varepsilon$ is a few dollars. Our method captures this difference. We see how there is a steep decrease in accuracy when the cost of the modification is low (under 1$) and thus the training needs to account for likely adversarial actions. Then, it enters a plateau when the cost of emails increases and no changes are possible given the adversary's budget. When the budget $\varepsilon$ grows enough to buy more expensive emails, the accuracy decreases again. We also observe that the gain in clean accuracy is higher than for robust accuracy. This effect is due to a better representation of the underlying categorical data distribution.

**Universal Robust Embeddings.** In order to evaluate our Bilevel alternating minimization framework, we run Algorithm 2 on TabNet to produce robust embeddings, and we merge these embeddings using Algorithm 3. After that, we attack the model using budgets $\varepsilon$ set to 10$ for IEEECIS, 1$ for BAF, and 100$ for Credit. These budgets enable the adversary to perform most of the possible transformations. We compare our methods on gradient-boosted stumps because training gradient boosted decision-trees with the method of Wang et al. (2020) is computationally infeasible for an adequate number of estimators and high dimensional data.

We show the results in Table 1. In almost all setups, our method yields a significant increase in robustness. For example, for IEEECIS we increase the robust accuracy of LGBM by 24%, obtaining a model that combines both high clean and robust accuracy, and that outperforms TabNet. Our method outperforms Wang et al. (2020) both for clean and robust accuracy. These results confirm that training based on only $l_1$ distance is not sufficient for tabular data .

We also compare the performance of the different methods with respect to the time spent on training and merging (see Table 2). We see that our method is considerably faster than previous work. The improvement is tremendous for IEEECIS and Credit where the dimensionality of the input is more than 200 and the number of estimators are 80 and 100 respectively. In the Appendix C, we also discuss how robust embeddings can be applied to a neural network of the same type.

| Dataset | Training RE | GBS-RE | GBS-W |
|---------|-------------|--------|-------|
| IEEECIS | 9.5 | 0.06 | 233.3 |
| BAF | 3.17 | 0.02 | 4.97 |
| Credit | 3.24 | 0.07 | 174.7 |

Table 2: **Computation time.** Time is measured in minutes. The total time of training RE and training GBS-RE is less than Wang et al. (2020)'s training

## 6 Conclusion

In this work, we propose methods to improve the robustness of models trained on categorical data both for deep networks and tree-based models. We construct a benchmark with datasets that enable

robustness evaluation considering the financial constraints of a real-world adversary. Using this benchmark, we empirically show that our methods outperform previous work while providing a significant gain in efficiency.

**Limitations and Future Work.** In our work, we focused on development of a method to improve the robustness of a machine learning model, having as small degradation in performance as possible. However, there are applications where even a small accuracy drop would incur more financial losses than potential adversarial behaviour. These setups can be still not appropriate for our framework. Quantifying these trade-off can be a promising direction for future work, and one way of doing it would be to follow a utility-based approach introduced in Kireev et al. (2022). Besides that, we do not cover the extreme cases where the dataset is either too small and causes overfitting or too large and makes adversarial training more expensive. Addressing these extreme cases can be also considered as a research direction. Finally, we leave out of the scope potential problems in the estimation of adversarial capability, e.g., if the cost model which we assume is wrong and how it can affect both the robustness and utility of our system.

# Acknowledgements

M.A. was supported by the Google Fellowship and Open Phil AI Fellowship.

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

# A   Cost modelling and data preprocessing

For all datasets used in the evaluation, we perform threat modelling based on the information about possible perturbations. In this section, we describe how we assign cost matrices to modifications of the features in each dataset and the common methods which we use for data preprocessing. Note that in this section for some features we mostly provide the methods and examples, and all the exact values are listed in the source code (`exp/loaders.py`). It is also worth mentioning that the exact prices might fluctuate, and our goal is to provide an estimate to demonstrate the capabilities of our methods.

## A.1   Data Preprocessing

All the datasets were balanced for training using random undersampling. Numerical features were binned in 10 bins since a further increase in the number of bins gave no additional improvement. In order to avoid numerical issues, we set the minimal possible cost to 0.1\$, and the maximal possible cost to 10000\$ (100 times greater than the largest $\varepsilon$ used in the evaluation). For the attack, we assume that the adversary is attacking only for one target class (e.g. their aim is to get a transaction marked as "non-fraud").

## A.2   `IEEECIS`

In evaluation on `IEEECIS` we use the following features for our perturbation set:

- *P_emaildomain* - Email domain which can be used by an adversary. We assume that an adversary can either buy a new email or create it themself. In the first case we take the existing prices from dedicated online shops (e.g. adversary can buy *\*@yahoo.com* email for 0.12\$ at *buyaccs.com*). In the second case, we estimate minimal expenditures to register one. For example, some emails require buying services of an internet provider (*\*@earthlink.net*), in this case, we estimate the minimal price of such service and count it as an adversarial cost (50\$ for *\*@earthlink.net*).

- *DeviceType* - Device type represents the operating system installed on the adversary's device. This feature can be easily manipulated since the adversary controls the device and can change device information sent to the system or use an emulator. We ascribe minimal cost (0.1\$) to this type of perturbation.

- *Card_type* - Card type is a brand of a credit or debit card used by the adversary. For this feature, we assume that the adversary needs to buy a stolen credit or debit card. We estimate the cost of such purchase to be 20\$ for Visa or MasterCard and 25\$ for less common Discovery or American Express. This estimation is based on stolen card prices in the darknet online marketplaces.

## A.3   `BAF`

In the evaluation on the `BAF` dataset we use the following features for our perturbation set:

- *income* - Income of a credit applicant. This feature can be perturbed by buying a fake paystub, which costs around 10\$.

- *zip_count_4w* - Number of applicants with the same zip code. The adversary can change their address by paying just around 1\$ (In US).

- *Device-related features* - Here we consider features which are related to the device/OS used by an adversary, and which can be manipulated the same way as *DeviceType* in `IEEECIS`. *foreign_request, source, keep_alive_session, device_os* belongs to this feature type. The cost model for these features is the same as for *DeviceType*.

- *email_is_free* - This feature represents if the adversary uses a paid email or a free one. Since going from paid to free is cheap, we represent this transition with the smallest price of 0.1\$. For going from a free email to the paid one we use the cheapest paid email from `IEEECIS`, and set this cost to 10\$.

- *phone_home_valid, phone_mobile_valid* - Validity of the provided phone number. The logic is the same as with the previous feature. It costs almost nothing (0.1$) to change from valid to non-valid, and 10$ vice versa.
- *device_distinct_emails_8w* - Number of distinct email addresses used by the adversary. This feature can only be increased since the adversary presumably cannot delete its records from the system. The cost for an increase is small (0.1$), decreasing is impossible.

### A.4 `Credit`

Finally, for the `Credit` dataset we use:

- *Merchant City* - Location where the purchase was made. Without any prior knowledge, we assume that this feature can only be manipulated physically by moving to a different location. Therefore, in this case, the cost of change is the cost of a transport ticket to this location. As the lower bound, we use a transport price of 0.1$ per km of distance between two cities. Therefore, the total cost of changing this feature from city A to city B, we estimate as the distance from A to B, multiplied by 0.1$.
- *card_type, card_brand* - For these 2 features we use the same cost model as for the *Card_type* feature in `IEEECIS` dataset.

## B  Additional details on experiments

### B.1  Experimental setup

All the final experiments were done on AMD Ryzen 4750G CPU, Nvidia RTX 3070 GPU, and Ubuntu 22.04 OS.

### B.2  Hyperparameters

We list our evaluation parameters in Table 1. The TabNet parameters are denoted according to the original paper Arik & Pfister (2019). We set the virtual batch size to 512. Most of the hyperparameters were selected via a grid search.

During adversarial training with a large $\varepsilon$, we encountered catastrophic overfitting on `IEEECIS` and `Credit`. For example during adversarial training with $\varepsilon = 100$ on `IEEECIS` we observe the overfitting pattern in test robust accuracy (Table 2). In these situations, we reduced the number of epochs.

## C  Additional experiments

In this section, we show some additional experiments clarifying our design decisions and describing some properties of universal robust embeddings.

### C.1  Comparison of first and last layer embeddings

As we mentioned in our paper, we only consider the first-layer embeddings of the neural network. The main justification for this design choice is that for many tabular datasets, "classical" methods like

| Model | Normal Training | FL Embeddings | LL Embeddings |
|-------|-----------------|---------------|---------------|
| LGBM  | 81.2            | 81.0          | 78.9          |
| RF    | 83.6            | 83.0          | 78.8          |

Table 3: **Standard accuracy for the first vs. last layer embeddings.** The first layer embeddings (FL) consistently outperform the last layer embeddings (LL).

LGBM or Random Forest can outperform neural networks. A good example of this case is the `IEEECIS` dataset. Both in the original Kaggle competition and in our experiments, decision trees show better performance than neural networks. To demonstrate this effect we run a simple experiment where we pass the last-layer embeddings to the LGBM and RF and compare the performance with

Table 1: **Hyperparameters used in the evaluation**

| Parameter | Value range |
| --- | --- |
| `All Datasets` | |
| $\tau$ | 0.9 |
| *Dykstra iterations* | 20 |
| `IEEECIS` | |
| *Batch size* | 2048 |
| *Number of epochs* | 400 |
| *PGD iteration number* | 20 |
| *TabNet hyperparameters* | $N_D = 32, N_A = 32, N_{steps} = 4$ |
| $\varepsilon$, *$* | $[0.1, 0.2, 0.3, 0.5, 1.0, 3.0, 5.0, 10.0, 30.0, 50.0, 100.0]$ |
| `BAF` | |
| *Batch size* | 1024 |
| *Number of epochs* | 30 |
| *PGD iteration number* | 20 |
| *TabNet hyperparameters* | $N_D = 16, N_A = 16, N_{steps} = 3$ |
| $\varepsilon$, *$* | $[0.1, 0.2, 0.3, 0.5, 1.0, 3.0, 10.0, 30.0]$ |
| `Credit` | |
| *Batch size* | 2048 |
| *Number of epochs* | 100 |
| *PGD iteration number* | 20 |
| *TabNet hyperparameters* | $N_D = 64, N_A = 64, N_{steps} = 2$ |
| $\varepsilon$, *x10$* | $[1.0, 10.0, 30.0, 100.0, 300.0]$ |

Table 2: **Overfitting behaviour**

| Epoch | 1 | 5 | 10 | 15 | 20 | 30 |
| --- | --- | --- | --- | --- | --- | --- |
| *Test Robust Accuracy* | 51.5 | 52.4 | 55.0 | 61.2 | 53.4 | 52.1 |

normal LGBM/RF training. We report the results in Table 3. The results are rather expected: due to the data processing inequality, we can expect information loss if the data is processed by an inferior classifier for this task (i.e., a neural network). Therefore, we can conclude that first-layer embeddings are a better choice if we want to preserve the standard accuracy of the decision-tree based classifier.

## C.2 Influence of the embedding merging algorithm

Another topic which we want to cover in this section is the effect of the embedding merging algorithm. In this subsection, we would like to answer two questions:

1. *Do we need the merging algorithm?*

2. *If yes, does it work without properly trained embeddings?*

We can answer both questions using Table 4. We evaluate robust embeddings both with normal adversarial training and bilevel optimization for different values of the parameter $\tau$. For comparison, we also apply the merging algorithm to randomly generated embedding, in order to perform the ablation study for the effect of merging. We see that even merging with small $\tau$, significantly improves the robustness, keeping the clean accuracy on a high level for robust embeddings, while for random embeddings it has no effect. Also, we see that, although normal adversarial training has some effect on robust accuracy after embedding transfer, the bilevel optimization algorithm provides the models with a much better robustness accuracy trade-off.

Table 4: **Evaluation of the embedding merging algorithm.** We report clean and robust accuracy of boosted decision stump classifier on BAF dataset (in percentage) with both random and robust embeddings. For the robust embeddings we evaluate both normal adversarial training, and the bilevel minimization algorithm. $\tau = 0.0$ means that no merging is performed.

| Method | $\tau$ | | | | | |
| --- | --- | --- | --- | --- | --- | --- |
| | 0.0 | 0.02 | 0.05 | 0.1 | 0.15 | 0.2 |
| Robust Emb. (Bilevel), Clean | 74.9 | 74.5 | 74.1 | 71.9 | 71.6 | 68.1 |
| Robust Emb. (Bilevel), Robust | 48.3 | 48.3 | **51.9** | **61.4** | **70.9** | **67.7** |
| Robust Emb. (Normal AT), Clean | 74.3 | 73.5 | 72.4 | 71.8 | 70.8 | 70.8 |
| Robust Emb. (Normal AT), Robust | 48.3 | **51.5** | 51.0 | 55.1 | 53.3 | 53.4 |
| Random Embeddings, Clean | 74.5 | 71.9 | 71.7 | 71.7 | 71.7 | 68.8 |
| Random Embeddings, Robust | **49.1** | 44.8 | 44.7 | 44.7 | 44.7 | 47.3 |

## C.3 Experiments on FT transformer

In order to show that our approach is not tied to one particular neural network architecture, we performed the same experiments with the more recent *FT-Transformer* model (Gorishniy et al., 2021). We applied robust embeddings generated for FT-Transformer to the models used in Section 5. The results are shown in Table 5. Overall, we see that FT-Transformer embeddings give better clean and robust performance, and therefore we conclude that our method is also suitable for this type of neural networks.

Table 5: **Robust embeddings for FT-Transformer.** We evaluate robust embeddings generated with FT-Transformer (RE-FT) (Gorishniy et al., 2021), in the same scenario as for the Table 1.

| Model | No Emb. | RE | RE-FT |
| --- | --- | --- | --- |
| RF, *Clean* | 72.3 | 65.8 | 67.5 |
| RF, *Robust* | 42.8 | 65.8 | 67.1 |
| GBS, *Clean* | 74.1 | 68.1 | 71.5 |
| GBS, *Robust* | 46.8 | 67.7 | 71.5 |
| LGBM, *Clean* | 74.1 | 68.1 | 71.4 |
| LGBM, *Robust* | 49.2 | 67.5 | 71.4 |
| CB, *Clean* | 74.4 | 67.2 | 70.8 |
| CB, *Robust* | 48.2 | 67.2 | 70.8 |

## C.4 Effect of embedding transfer for neural networks

In this section, we cover the effect of robust embeddings on the original neural network trained over the clean data. To measure this effect we apply robust embeddings to a TabNet model trained on clean data. The results are reported in Table 6. It shows that the target neural network definitely benefits from this procedure, though the resulting models still have less robustness than models trained with CatPGD 1. We do not propose this technique as an optimal defence since Cat-PGD is available for neural networks, however, it can provide some robustness if adversarial training is not available for some reason (for example due to computational recourses limitation).

Table 6: **Embedding Transfer.** We evaluate TabNet models trained on clean data with robust embeddings applied to them, in the same setting as we do for the Table 1. We compare its performance with clean models (NN) and models trained with CatPGD (NN-CatPGD).

| Model | NN | NN-RE | NN-CatPGD |
|---|---|---|---|
| IEEECIS, *Clean* | 79.0 | 69.0 | 74.9 |
| IEEECIS, *Robust* | 40.1 | 61.9 | 72.6 |
| BAF, *Clean* | 75.5 | 69.9 | 70.9 |
| BAF, *Robust* | 48.3 | 67.9 | 70.4 |
| Credit, *Clean* | 83.1 | 82.1 | 72.8 |
| Credit, *Robust* | 70.6 | 71.5 | 72.6 |

