# OpenReview forum: "Transferable Adversarial Robustness for Categorical Data via Universal Robust Embeddings"
_NeurIPS.cc/2023/Conference — NeurIPS 2023 poster_

### Official Review · Reviewer_EHdN · 2023-07-04

**Soundness:** 3 good
**Presentation:** 2 fair
**Contribution:** 3 good
**Rating:** 5
**Confidence:** 3

**Summary:**

This paper proposes to use the input embeddings from neural networks as universal robust embeddings. They can be used to enhance the robustness of robustness of machine learning models.

**Strengths:**

This paper is well written and well formulated.

**Weaknesses:**

- It is better to add more theoretical analysis about why embeddings from neural networks still work on other machine learning models.
- Minor writing issues:
    - "Robustness for models other than neural networks" should be "Robustness for models other than neural networks. " in Line 49
    -  "asdecision trees", in Line 64
    -  ”virtual world” with a ”virtual population”, in Line 277


**Questions:**

- Universal Robust Embeddings come from adversarial neural network models while they are not tested on them. Do Universal Robust Embeddings keep useful in neural network models?
- How does the embedding for each cluster generated in Algorithm 3?



**Limitations:**

No.

---

> ### Author Rebuttal · Authors · 2023-08-09
>
> We thank you for your comments.
>
> > *It is better to add more theoretical analysis about why embeddings from neural networks still work on other machine learning models.*
>
> We agree with this suggestion. We mentioned the potential reasons behind this in Lines 197-200: *“By using first layer embeddings, we avoid a potential loss of information (unless the embedding matrix Q has exactly identical rows which are unlikely in practice) that would occur if we were to use embeddings from the final layer.”* We can formalize this statement as a proposition and include a proof in the appendix. At a high level, if the embedding matrix does not contain completely identical embeddings before merging, any potential split that existed before the embedding process is preserved if the threshold is sufficiently low.
>
> > Minor writing issues:
> >- "Robustness for models other than neural networks" should be "Robustness for models other than neural networks. " in Line 49
> >- "asdecision trees", in Line 64
> >- ”virtual world” with a ”virtual population”, in Line 277
>
> We thank you for carefully reading our paper. We will definitely fix these typos in the revised version of the paper.
>
> > *Universal Robust Embeddings come from adversarial neural network models while they are not tested on them. Do Universal Robust Embeddings keep useful in neural network models?*
>
> Universal robust embeddings should also lead to substantial robustness in neural networks. However, since one can directly train robust neural networks *end-to-end* using Cat-PGD (described in Algorithm 1), we did not include such results in the paper. To answer your question we performed an experiment using robust embedding for a clean neural network similar to the experiments in Table 1. The results are the following:
>
> | _Dataset_        | NN | NN-RE |  NN-CatPGD |
> |------------------|:----------:|:-------------:| :----------------:|
> | IEEECIS _Clean_  | 79.0     | 69.0        | 74.9 |
> | IEEECIS _Robust_ | 40.1     | 61.9        |  72.6 |
> | BAF _Clean_      | 75.5     | 69.9        | 70.9  |
> | BAF _Robust_     | 48.3    | 67.9        | 70.4 |
> | Credit _Clean_   | 83.1     | 82.1        | 72.8  |
> | Credit _Robust_  | 70.6     | 71.5        | 72.6 |
>
>
> These results show that the target neural network definitely benefits from this procedure, though the resulting models still have less robustness than models trained with CatPGD (Figure 1). We will include this experiment in the revised version of the paper to support the idea of *universality* of our robust embeddings.
>
> > *How does the embedding for each cluster generated in Algorithm 3?*
>
> At the end of the algorithm, we assign the **center** of each cluster as a separate embedding (we compute the center as the standard mean of all vectors belonging to a cluster). We will clarify this step in Algorithm 3.

---

### Official Review · Reviewer_9Bah · 2023-07-07

**Soundness:** 3 good
**Presentation:** 3 good
**Contribution:** 3 good
**Rating:** 6
**Confidence:** 4

**Summary:**

This paper investigates adversarial robustness of tabular data. By transferring embeddings achieved by bilevel alternating minimization framework, tree-based models can then learn robustness without the need of adversarial training.

**Strengths:**

The proposed approach is novel and seems to be general for wide application with categorical features.

The robustness is considered under the financial constraints which is very practical.

Empirical results validate the good performance of the proposed approach.

The ablation study in the appendix is interesting and insightful.

**Weaknesses:**

In line 172 “There is currently no existing algorithm in the literature that enables us to operate within this cost-based objective for decision-tree based classifiers.” The claim seems not precise. Chen et al. (2021) has defined a similar cost-aware threat model, gained robustness for tree models, and can support categorical data. So it should serve as a descent baseline for the proposed approach. First, comparing to Chen et al. (2021), the paper should elaborate more details regarding the threat model and training approach differences in related work. Current sentences in related work are unclear for differences. Empirical comparisons for both robustness and efficiency should be provided to make the difference more clear. I am happy to raise the score if this concern is addressed.

**Questions:**

The paper focuses on tabular data. However, the proposed approach seems to fit the mixture of any categorical and numerical data. A discussion regarding the limitations/scope of the approach would be beneficial.

**Limitations:**

No negative societal concern

---

> ### Author Rebuttal · Authors · 2023-08-09
>
> We thank you for pointing out the novelty and practicality of our approach. We omitted the comparison with Chen et al. (2021) in our work, since we consider its threat model to be sufficiently different. The two key differences are:
>
> 1. Chen et al. operate within per-feature constraints, while we argue that a realistic adversary is bounded by a total per-sample budget. At a high level, the difference here is similar to the difference between $\ell_1$ and $\ell_\infty$ sets. Let us consider an ML problem with 100-dimensional feature input (for example multiple categorical variables) where some of the features can be changed for \\$100 and some for \\$10. Assume an adversary willing to spend \\$150. If we use Chen et al’s per-feature constraint we need to either prohibit all feature changes (since an adversary can change any of them) and lose all utility; or permit all of them,losing all robustness. This situation arises because in Chen et al. model there is no difference between modifying 1 or 100 features. If we take a per-sample budget as in our approach, the adversary would not be able to modify more than 15 features (\\$15*10) or even less if the total number of categorical features is less than 15.
>
> 2. The constraints of Chen et al. are assumed to be symmetrical and uniform for categorical features (supporting other models would require significant modifications of their algorithm) without preserving the feature space. In its current form, Chen et al.’s algorithm does not work with categorical features (and in their evaluation, they use datasets with numerical features). In theory, we could apply it to categorical features if we first encoded them and then set per-feature constraints. However, such encoding would need to account for the cost model. Example: the adversary is capped at \\$200 and the categorical feature has four values: New York, Boston, Chicago, Istanbul. Assume that if the starting value is Istanbul, all changes should be prohibited (flights to the US cost more than \\$200). If it is New York, only Istanbul should be prohibited. In Chen et al. this could not be supported without significant modifications to the algorithm, thus the feature constraint must be the same for all samples.
>
> Due to these reasons, we chose Wang et al. (2020) as a baseline since their proposed $\ell_1$ constraint model is closer to our setting.
> However, for the sake of completeness, we have performed an empirical comparison to the method of Chen et al. during the rebuttal time. To this end, we have done a best-effort adaptation of Chen et al’s code to address the issues above, and we compared it with our method on random forest (since this classifier was used in both our and their evaluation):
>
>  | Dataset   | Ours | Chen et al. |
> |---------------|------|---------|
> | IEEECIS, _Clean_ | 81.0 | 69.0 |
> | IEEECIS, _Robust_ | 81.0 | 69.0 |
> | BAF, _Clean_ | 65.8 |  61.3 |
> | BAF, _Robust_ | 65.8 |  61.3 |
>
> We could not compare to the $\texttt{Credit}$ dataset as Chen et al’s published code does not work well with the high-dimensional input of this dataset and thus its execution could not finish until the rebuttal deadline.
>
> As both our and Chen et al. pipelines produce the same model if no robust training is applied, we conclude that we did our best for the comparison, and our approach significantly outperforms Chen et al.’s  within our financial threat model.
>
> > *The paper focuses on tabular data. However, the proposed approach seems to fit the mixture of any categorical and numerical data. A discussion regarding the limitations/scope of the approach would be beneficial.*
>
> We thank you for this suggestion. Unfortunately, there is no clear definition in the literature of what tabular data comprises. In our paper, we take the definition of *“a mixture of categorical and numerical data”*. We can highlight this better to ensure that the scope of our methods is well-defined.

---

> > ### Comment · Reviewer_9Bah · 2023-08-13
> > **Thanks for the response**
> >
> > Thanks to the authors for the clarification and additional comparisons especially for comparisons to Chen et al.'s, which have addressed my concerns. Hope the authors can reflect the clarifications and results in the camera ready. As promised, I raised my score to 6.

---

> > > ### Author Response · Authors · 2023-08-19
> > > **Thank you for your response**
> > >
> > > We thank you for updating your score. We will definitely include these results in the final version of the paper.

---

### Official Review · Reviewer_9uXH · 2023-07-07

**Soundness:** 3 good
**Presentation:** 3 good
**Contribution:** 3 good
**Rating:** 5
**Confidence:** 3

**Summary:**

This paper presents a method to train adversarially robust deep networks for tabular data and to transfer this robustness to other classifiers via universal robust embeddings tailored to categorical data. These embeddings created using a bilevel alternating minimization framework can be transferred to boosted trees or random forests making them robust without the need for adversarial training while preserving their high accuracy on tabular data. This paper shows that the proposed methods outperform existing techniques within a practical threat model suitable for tabular data.

**Strengths:**

1. This paper proposes a practical adversarial training algorithm supporting complex and heterogeneous constraints for categorical data that can accurately reﬂect financial costs for the adversary. The proposed training algorithm is based on the continuous relaxation of a discrete optimization problem and employs approaches from projections onto an intersection of convex sets.
2. This paper proposes a method to generate universal robust embeddings, which can be used for transferring robustness from neural networks to other types of machine learning models such as decision trees or random forests. This paper uses existing datasets to build the first benchmark to evaluate robustness for tabular tasks in which the adversary is constrained by financial capabilities.


**Weaknesses:**

1. There is currently no existing algorithm in the literature to operate within this cost-based objective for decision-tree based classifiers. Yet, it would be easier to understand the proposed approach if this paper could provide more explanations about the difference of using gradient-based adversarial training on the tabular data rather than the image or text data.
2. During adversarial training with a large budget, this paper encountered the catastrophic overftting on IEEECIS and Credit. The authors reduced the number of epochs in these situations. It would be more convincing if the authors could present the empirical evidences and the related analysis of the catastrophic overftting in this paper for the tabular data.
3. This paper claims that most of the hyperparameters used in the evaluation were selected via a grid search. Yet, it could be more sufficient if this paper could provide the performance of the proposed approach with different hyperparameters to further analyze the threat modeling for tabular data.


**Questions:**

This paper selects three publicly-available datasets that fit the criteria and all datasets were balanced for the provided experiments. Yet, the datasets for the tabular data to evaluate the adversarial robustness are not that common compared with the dataset of image and text. It could be more persuasive if the authors provide further clarifications for the standards and details to process the tabular data.

**Limitations:**

This paper presents a method to train adversarially robust deep networks for tabular data and construct a benchmark with datasets. Nevertheless, it could be more sufficient with improvements. The detailed comments can be seen in the weaknesses.

---

> ### Author Rebuttal · Authors · 2023-08-09
>
> We thank the reviewer for the positive assessment of our work.
>
> > *Yet, it would be easier to understand the proposed approach if this paper could provide more explanations about the difference of using gradient-based adversarial training on the tabular data rather than the image or text data.*
>
> There are two main differences:
> (a) the perturbation set is inherently discrete for categorical data (contrary, e.g., to perturbations for images which are typically modeled as continuous perturbation sets) and
> (b) categorical data changes have an associated (financial) cost as opposed to modifying images where it is hard to quantify a cost model for pixel changes. Existing methods do not support cost-based constraints.
>
> Thus, new methods tailored to tabular data and the tabular adversary constraints are needed (such as our Cat-PGD described in Section 4.1). Our paper also points out that the tabular scenario requires careful design of the optimization problem to enable efficient computations. For example, our method relies on a tight relaxation of the problem to the continuous domain; and uses non-differentiable tree-based models which are not widely used for text data and require new techniques to handle.
> We will make these points clearer in the revised version of the paper.
>
> > *During adversarial training with a large budget, this paper encountered the catastrophic overftting on IEEECIS and Credit. The authors reduced the number of epochs in these situations. It would be more convincing if the authors could present the empirical evidences and the related analysis of the catastrophic overftting in this paper for the tabular data.*
>
> We agree this is an important practical issue, and apologize for not having included initially the empirical evidence we gathered during our experiments. We will add such analysis in the revised version of the paper.
>
> Shortly, in our experiments we noticed that, when trained with large budgets, neural networks overfit: the robust accuracy measured using the discrete graph-search-based attack started to drop after a certain epoch. For example for $\varepsilon=100$ on IEEECIS trained with 10 PGD iterations, we observed the following process:
>
> | Epoch                | 1    | 5    | 10   | 15   | 20   | 30   |
> |----------------------|------|------|------|------|------|------|
> | Test Robust Accuracy | 51.5 | 52.4 | 55.0 | 61.2 | 53.4 | 52.1 |
>
> After 20 epochs, the test robust accuracy suddenly gets degraded (as in catastrophic overfitting), and drops to the level of a clean model (i.e., close to $50\\%$).
>
>
> To address the issue, we applied measures proposed in the catastrophic overfitting literature, e.g.,  increasing the number of PGD iterations or initializating with Gaussian noise. These measures  drastically improved the situation, eliminating the sudden drop. Yet, they did not completely eliminate the issue.  After 150 epochs, we still see a slight robust accuracy drop (only 2%).
> > *This paper claims that most of the hyperparameters used in the evaluation were selected via a grid search. Yet, it could be more sufficient if this paper could provide the performance of the proposed approach with different hyperparameters to further analyze the threat modeling for tabular data.*
>
> While we selected most hyperparameters via a grid search, we analyzed in detail the only  hyperparameter that is related to the threat modeling: the cost bound of the adversary $\varepsilon$. For example, we trained multiple models with different $\varepsilon$’s in Figure 1 to analyze the effect of the adversary’s budget on the tradeoff between robustness and accuracy. We are happy to include additional results along these lines if you have some particular suggestions.
>
> > *It could be more persuasive if the authors provide further clarifications for the standards and details to process the tabular data.*
>
> We provide these details in Appendix: **A.1 Data Preprocessing** contains details regarding general data preprocessing, and **A.2 IEEECIS**, **A.3 BAF**, **A.4 Credit** contain details regarding the costs associated with each categorical feature. We will include an explicit reference to this part of the appendix in the revised version of the paper. Also, we are open to add more details if something is still missing in our description of the data or preprocessing pipeline.

---

> > ### Comment · Reviewer_9uXH · 2023-08-19
> >
> > Thanks for the detailed response. The differences of the adversarial training on the tabular data and common data (such as image or text) have been clarified and explained by the authors. There are also some preliminary observations provided to verify the effectiveness of the proposed approach, which partly address my concerns. I would maintain the score.

---

### Official Review · Reviewer_a4Uk · 2023-07-09

**Soundness:** 2 fair
**Presentation:** 3 good
**Contribution:** 2 fair
**Rating:** 6
**Confidence:** 3

**Summary:**

This study presents novel approaches aimed at enhancing the robustness of models trained on categorical data on the deep networks and tree-based models. It addresses the challenges posed by real-world adversaries within the constraints of financial resources. Through extensive empirical analysis, they demonstrate that our proposed methods not only surpass prior research but also offer a substantial improvement in efficiency.

**Strengths:**

The paper is good-written and easy to follow.
The proposed method of universal robust embeddings is used for transferring robustness from neural networks to other types of machine learning models and obtains better robustness for tabular tasks in which the adversary is constrained by financial capabilities.




**Weaknesses:**

The proposed method is only applied to the financial dataset based on the meaningful cost of modifying features.
There are some confusion on the definition of this cost. Please refer to the questions.


**Questions:**

1. The assumption in line 138 is that the cost of modifying features is additive.
    If the two features have the inverse function for the final decision, is it reasonable to use this assumption?

2.  What parameters will affect the cost bound for different datasets?

3. Table 3 should be Table 2?

**Limitations:**

The paper didn't discuss the limitations of their work.

---

> ### Author Rebuttal · Authors · 2023-08-09
>
> We thank you for the positive feedback.
>
> > *1. The assumption in line 138 is that the cost of modifying features is additive. If the two features have the inverse function for the final decision, is it reasonable to use this assumption?*
>
> Yes, we are confident that it is a reasonable assumption since it is the most natural one. E.g., on the IEEECIS dataset, the attacker can change the following features: *email domain*, *device type*, and *card type*. Changing a feature incurs a certain cost for the adversary (for example buying an email address) and if the attacker wants to change multiple of them, the costs will add up. Moreover, even if the two features have the inverse function on the final decision, the adversary can change them to the “opposite” directions (to cause a misclassification) or choose not to change some of them at all if it is not profitable according to the total cost.
>
>
> > *2. What parameters will affect the cost bound for different datasets?*
>
> The cost bound depends on the financial capabilities of the attacker, i.e., on the amount that they are willing to spend to generate an adversarial example. It is completely up to the model owner to decide which cost bound is appropriate to represent a potential adversary. In all cases, we note that our method improves upon all cost bounds simultaneously compared to existing methods (see Figure 1).
>
>
> > *3. Table 3 should be Table 2?*
>
> Thanks for spotting this!  We will fix this in the revised version of the paper.

---

### Official Review · Reviewer_saBk · 2023-07-10

**Soundness:** 3 good
**Presentation:** 4 excellent
**Contribution:** 3 good
**Rating:** 6
**Confidence:** 4

**Summary:**

The authors propose an adversarial training algorithm for tabular data with categorical features which can produce robust embeddings for use with downstream non-differentiable models like decision trees.  The authors evaluate on three datasets and with several downstream models on the robust embeddings.

**Strengths:**

Adversarial training has generated massive numbers of papers but these overwhelmingly focus on image data, yet tabular data is more popular than image data in high-stakes industrial use-cases of ML.  Therefore, the potential for impact is high (at least insofar as adversarial robustness is impactful).  I also appreciate the attempt to enhance robustness for decision tree models too since they are still more popular than neural networks in industrial tabular ML.

The writing is also generally easy to understand.

**Weaknesses:**

My main concerns regard the thoroughness of experiments.

Gradient boosted decision tree methods like XGBoost and CatBoost often achieve superior performance compared to neural networks on many datasets which is one of the reasons they are still very popular.  Do they maintain their advantage over neural networks after adversarial training (both in terms of robustness and clean accuracy)?  If not, then there might not be a use-case for the robust embeddings.

Along the same lines, it would be good to use modern neural network architectures like FT Transformer or SAINT and also better non-neural models like XGBoost and CatBoost.

A major problem for adversarial defenses is gradient masking, or defense that might be hard to attack using certain attack algorithms but are not fundamentally robust.  The categorical variable setting in particular is especially difficult to attack, so it seems hard to know whether or not a model is truly robust.  You might consider trying other attacks as well to verify that the models are truly robust.  For example, you could try prompt tuning optimization strategies like PEZ (from “Hard Prompts Made Easy: Gradient-Based Discrete Optimization for Prompt Tuning and Discovery”).

Adversarial training is known to overfit, especially on small datasets.  It might be worth trying out the proposed algorithms on smaller datasets too.

Credit card fraud detection, in practice, is an area where every tiny tiny amount of accuracy is profitable.  Given the performance degradations involved here over non-robust models, it might be worthwhile to find application areas which are not so sensitive to clean accuracy as a motivating demonstration.

Minor: The conclusion is just a re-statement of the abstract.  Would be useful to put directions for future work.


**Questions:**

N/A

**Limitations:**

The authors did not discuss limitations in the main body as far as I can tell.

---

> ### Author Rebuttal · Authors · 2023-08-09
>
> We thank you for the detailed response. We discuss the raised concerns below.
> > *Gradient boosted decision tree methods like XGBoost and CatBoost often achieve superior performance compared to neural networks ... Do they maintain their advantage over neural networks after adversarial training (both in terms of robustness and clean accuracy)?*
>
> We agree that it is an important question. In the current literature, there is no clear consensus about which method is always superior: neural networks or gradient-boosted trees. Our results demonstrate a similar trend. Figure 1 and Table 1 show that for Credit and BAF datasets NN and LGBM performance is close, while for IEEECIS, TabNet has clearly superior performance even after robust embedding transfer. Therefore the answer is yes, tree-based classifiers can maintain the advantage and are a suitable use case for our robust embeddings.
>
> > *Along the same lines, it would be good to use modern neural network architectures like FT Transformer or SAINT and also better non-neural models like XGBoost and CatBoost.*
>
> We agree that the validation on a different neural network can benefit our ablation study. We incorporated the FT transformer into our pipeline to check if there is a significant difference. However, we did not manage to get better clean performance with the FT transformer on IEEECIS and Credit. Nonetheless, we did manage to get 0.5% clean improvement on BAF, and due to time constraints, we focused on this dataset. We performed the same experiments with robust embeddings and we report them in the table below. It shows that embeddings trained with FT transformers demonstrate better performance compared with TabNet embeddings. Possibly it is better due to improved clean performance. We will include these results in the revised version of the paper.
> | _Model_  | No Emb. | RE | RE-FT |
> |--------|:------:|:------:|:-------:|
> | RF, _Clean_  | 72.3 | 65.8 | 67.5 |
> | RF, _Robust_ | 42.8 | 65.8 | 67.1 |
> | GBS, _Clean_ | 74.1 | 68.1 | 71.5 |
> | GBS, _Robust_ | 46.8 | 67.7 | 71.5 |
> | LGBM, _Clean_  | 74.1 | 68.1 | 71.4 |
> | LGBM, _Robust_ | 49.2 | 67.5 | 71.4 |
> | CatBoost, _Clean_  | 74.4 | 67.2 | 70.8 |
> | CatBoost, _Robust_ | 48.2 | 67.2 | 70.8 |
>
> In the paper, we performed experiments with LightGBM, a recent alternative (2017) which performs similarly or slightly better than XGBoost (2014). We thank you for pointing out CatBoost, as it is more recent than the gradient-boosted stumps (GBS) approach we evaluated. We performed the evaluation for this model (see table below, where the robust version is marked as -RE). The results are similar to other models, though it clearly performs better than basic GBS. We will include these results in the revised version of the paper.
> | _Dataset_   | CatBoost | CatBoost-RE |
> |--------|:------:|:------:|
> | IEEECIS _Clean_ | 76.5 | 76.1 |
> | IEEECIS _Robust_ | 51.1 | 72.0 |
> | BAF _Clean_ | 74.4 | 67.2 |
> | BAF _Robust_ | 48.2 | 67.2 |
> | Credit _Clean_ | 83.3 | 79.7 |
> | Credit _Robust_  | 71.7 | 71.9 |
>
>
>
> > *A major problem for adversarial defenses is gradient masking, or defense that might be hard to attack using certain attack algorithms but are not fundamentally robust. …You might consider trying other attacks as well to verify that the models are truly robust.*
>
> First, we would like to note that the attack we use in our evaluation is not gradient-based. This rules out the possibility of gradient masking as defined in the original paper [Practical Black-Box Attacks against Machine Learning](https://arxiv.org/abs/1602.02697).
>
> Instead, our attack attempts to solve the discrete optimization problem directly using a graph search method. Of course, it is always possible that our attack is suboptimal since the underlying problem is NP-hard. In order to ensure that we are not underestimating the adversary, we compare it to the performance of Uniform Cost Search (UCS). UCS does not rely on any heuristic. It performs an exhaustive search and always finds an optimal solution within its cost budget. Since UCS is very time-consuming, we could only run it on IEEECIS models. For all cases, we find that the difference in attack success between our attack and UCS is less than 1%. These results are in line with the findings of Kireev et al. (2022) (see Figure 1 in their paper). Since UCS is theoretically optimal, we conclude that there is not much room for improvement beyond the results of our graph search attack algorithm and thus our results do not underestimate the adversary’s power and can be taken as a reliable measurement of the robustness of the models.
>
> > *Adversarial training is known to overfit, especially on small datasets. It might be worth trying out the proposed algorithms on smaller datasets too.*
>
> We thank the reviewer for the comment. In our evaluation, we used the datasets on different scales (100K - 1M), and to avoid overfitting we grid-searched the capacity parameters of the Neural Network for each dataset. In the revised version of the paper, we will include the experiments on reduced versions of the dataset, and if our method is less effective there, we will include a discussion on this to the limitations section.
>
> > *Given the performance degradations involved here over non-robust models, it might be worthwhile to find application areas which are not so sensitive to clean accuracy as a motivating demonstration.*
>
> We agree that the decrease in clean accuracy is a concern. However, it is also up to the model owner to decide which model to deploy to maximize profits while reducing potential losses from fraudulent transactions. This is precisely the goal of Figure 1: to provide information about the trade-offs with respect to different cost bounds.
>
> We would also like to note that fraud detection is of particular interest in the adversarial context precisely because they are linked to financial gains, both on the side of the model owner *and* adversary. This is why we selected them for our evaluation.

---

> > ### Comment · Reviewer_saBk · 2023-08-13
> > **Thanks for the response**
> >
> > I thank the authors for adding CatBoost and modern tabular neural networks (at least one so far).  TabNet is widely agreed upon to be quite a bad model for tabular data compared to more recent architectures.
> >
> > I also want to push back against the applicability of this work to cc fraud detection.  Small accuracy differences in the second decimal place are worth astronomical amounts of money for a provider which processes unthinkable numbers of transactions per minute, so the sacrifices required for adversarial robustness are simply impractical.  This tradeoff is one of the major reasons why industrial practitioners largely don't care much about adversarial robustness.  It would be valuable for this work to highlight actual use-cases for your work since this is a method paper, and such use-cases are not obvious to me.
> >
> > I have raised my score due to your responses.

---

> > > ### Author Response · Authors · 2023-08-19
> > > **Thank you for your response**
> > >
> > > We thank you for updating your score. Concerning the drop in accuracy, it is an issue in all research on adversarial robustness. The question of which tradeoff would be acceptable to practitioners is an important one, unfortunately, it is hard to find data on the financial impact of adversarial behaviour. We will include a discussion on this concern in the paper to better contextualize the results.

---

### Author Rebuttal · Authors · 2023-08-09

We thank the reviewers for the feedback. We are glad that our work received positive comments such as:
- *“the potential for impact is high (at least insofar as adversarial robustness is impactful)”* (**Reviewer saBk**).
- *“The paper is good-written and easy to follow”* (**Reviewer a4Uk**)
- *”This paper proposes a practical adversarial training algorithm supporting complex and heterogeneous constraints for categorical data that can accurately reﬂect financial costs for the adversary.”* (**Reviewer 9uXH**)
- *“The proposed approach is novel and seems to be general for wide application with categorical features.”* (**Reviewer 9Bah**)
- *”This paper is well written and well formulated”* (**Reviewer EHdN**)

Following the suggestions by the reviewers, we have improved our empirical evaluation by adding extra experiments:
-  Comparison to the method of Chen et al. (2021)
-  CatBoost evaluation
-  Experiments on FT Transformer
-  Comparison of our attack with an exhaustive graph search
-  Catastrophic overfitting clarification
-  Experiments on transferring embeddings to a neural network model

We will carefully take into account all writing suggestions including:
- **Reviewer saBk**: add future directions to the conclusions.
- **Reviewer saBk** and **Reviewer a4Uk**: add a limitation section.
- Fix the typos pointed out by the reviewers.

We thank the reviewers again, and we are happy to engage in a follow-up discussion.

---

### Decision · Program_Chairs · 2023-09-21

**Decision:**

Accept (poster)

**Comment:**

This paper makes an important investigation into training adversarially robust models for tabular data. The paper also considers both neural models and decision trees in this setting. The reviewers are all positive about this work - the problem setting is important and less well-studied, the experimental investigation seems sound (all the concerns raised were sufficiently addressed). It would be great if the authors could add the additional experiments that came up in the reviews to the next version of this paper. Congratulations on the nice work!